# Factors Influencing the Prevalence of Resistance-Associated Substitutions in NS5A Protein in Treatment-Naive Patients with Chronic Hepatitis C

**DOI:** 10.3390/biomedicines8040080

**Published:** 2020-04-07

**Authors:** Karen K. Kyuregyan, Vera S. Kichatova, Anastasiya A. Karlsen, Olga V. Isaeva, Sergei A. Solonin, Stefan Petkov, Morten Nielsen, Maria G. Isaguliants, Mikhail I. Mikhailov

**Affiliations:** 1Russian Medical Academy of Continuous Professional Education, 125993 Moscow, Russia; 2Mechnikov Research Institute for Vaccines and Sera, 105064 Moscow, Russia; 3Chumakov Federal Scientific Center for Research and Development of Immune-and-Biological Products of Russian Academy of Sciences, 108819 Moscow, Russia; 4N.V. Sklifosovsky Research Institute for Emergency Medicine, 129090 Moscow, Russia; 5Department of Microbiology, Tumor and Cell Biology, Biomedicum, Karolinska Institute, 17165 Solna, Sweden; 6Instituto de Investigaciones Biotecnológicas, Universidad Nacional de San Martín, San Martín CP1650, Argentina; 7Department of Bio and Health Informatics, Technical University of Denmark, DK-2800 Kgs. Lyngby, Denmark; 8NF Gamaleja Research Center of Epidemiology and Microbiology, 123098 Moscow, Russia; 9Research Department, Riga Stradins University, LV-1007 Riga, Latvia

**Keywords:** hepatitis C virus (HCV), NS5A, resistance-associated substitutions, direct-acting antivirals, amino acid covariance, immune escape

## Abstract

Direct-acting antivirals (DAAs) revolutionized treatment of hepatitis C virus (HCV) infection. Resistance-associated substitutions (RASs) present at the baseline impair response to DAA due to rapid selection of resistant HCV strains. NS5A is indispensable target of the current DAA treatment regimens. We evaluated prevalence of RASs in NS5A in DAA-naïve patients infected with HCV 1a (*n* = 19), 1b (*n* = 93), and 3a (*n* = 90) before systematic DAA application in the territory of the Russian Federation. Total proportion of strains carrying at least one RAS constituted 35.1% (71/202). In HCV 1a we detected only M28V (57.9%) attributed to a founder effect. Common RASs in HCV 1b were R30Q (7.5%), L31M (5.4%), P58S (4.4%), and Y93H (5.4%); in HCV 3a, A30S (31.0%), A30K (5.7%), S62L (8.9%), and Y93H (2.2%). Prevalence of RASs in NS5A of HCV 1b and 3a was similar to that worldwide, including countries practicing massive DAA application, i.e., it was not related to treatment. NS5A with and without RASs exhibited different co-variance networks, which could be attributed to the necessity to preserve viral fitness. Majority of RASs were localized in polymorphic regions subjected to immune pressure, with selected substitutions allowing immune escape. Altogether, this explains high prevalence of RAS in NS5A and low barrier for their appearance in DAA-inexperienced population.

## 1. Introduction

The hepatitis C virus (HCV) is an infection of great societal impact, with high degree of chronicity resulting in liver fibrosis, cirrhosis, and eventually liver cancer, as well as extrahepatic manifestations such as cryoglobulinemia, insulin resistance, type 2 diabetes, neurological disorders, and lymphomas. Over the past several years, there has been a fundamental breakthrough in the treatment of chronic hepatitis C (CHC). Highly effective direct-acting antivirals (DAAs) have been developed and brought into use, targeting non-structural HCV proteins: NS3 protease, NS5B polymerase, and NS5A. The high effectiveness of DAAs turns them into a primary tool for cure of HCV infection and prevention of HCV-associated disorders and malignancies. Significant reduction in the incidence of this infection in the next 20 years can be expected, provided that this treatment is widely introduced [1].

The widespread use of DAAs raises the concern that these drugs may select for viral variants resistant to treatment. Although the problem of drug resistance for HCV is not as acute as it is for HIV, cases of failed HCV treatment due to the viral resistance to DAA have been repeatedly reported [2,3,4]. A number of polymorphisms in the viral genome/amino acid substitutions leading to resistance to DAA have been detected in NS3, NS5A, and NS5B proteins [5,6], but data on their prevalence among the general treatment naïve population is still limited. Such amino acid substitutions, reflecting polymorphisms in certain positions of NS3, NS5A, and NS5B, are referred to as a resistance-associated amino acid substitutions (RASs). The term “RAS” is used to refer to polymorphisms in viral genome in two occasions: (1) at baseline in treatment-naïve patients, and (2) for patients previously exposed to DAAs [4,5,6,7]. This study involved only the first category. The greatest clinical impact was demonstrated for the amino acid substitutions causing resistance to inhibitors of NS5A, which are the first line drugs for treatment of CHC (daclatasvir, ledipasvir, ombitasvir, and elbasvir) [3,7,8,9]. Russia has one of the world’s highest absolute number of HCV infections, and accounts, together with Egypt, China, India, Nigeria, and Pakistan, for more than half of the global HCV burden [10]. Up to 5 million people are estimated to be infected, with the most prevalent HCV genotypes being 1b and 3a [11,12]. By 2017, Russia had licensed CHC treatment with a NS5A inhibitor, daclatasvir, to be used in combination with the NS3 inhibitor asunaprevir or NS5B inhibitor sofosbuvir [13]. Now, when these DAAs are being introduced in the clinical practice, we need data on the prevalence of HCV strains with substitutions conferring resistance to DAA among treatment-naïve patients in the territory of the Russian Federation, to estimate treatment effectiveness, degree of non-response and help to select optimal treatment regimens.

NS5A, a ~450 amino acid multi-functional phosphoprotein, is an indispensable component of viral replication machinery [14]. NS5A-targeting DAA have complex effects on HCV replication. They interfere with RNA-binding [15], formation of protein–lipid complexes, and complex membrane associations [16]. This perturbs the delivery of viral genomes to replication complexes (clusters of structural and non-structural HCV proteins core, E2, NS4B, and NS5A) without altering HCV RNA colocalization with NS5A and viral egress from the cells [16]. The exact mechanism of action for NS5A inhibitors is not clearly understood. In vitro studies suggest that NS5A inhibitors reduce viral RNA production in new, rather than preformed, replication complexes, and also significantly shift the distribution of NS5A from the ER to lipid droplets [17]. Ascher DB et al. hypothesized that NS5A inhibitors act by favoring formation of dimeric structure(s) which do not bind RNA [15]. 

The purpose of this study was to determine the prevalence of RASs in NS5A protein and factors influencing their spread in treatment-naïve CHC patients in the Russian Federation before systematic application of DAA (which excludes circulation of de novo generated resistant strains). Majority of the detected RASs localized in the polymorphic regions of NS5A, some co-varied with specific amino acid residues not associated with resistance to DAA, possibly, to preserve viral fitness. Interestingly, we found these polymorphic regions to coincide with regions predicted and proven (by IEDB search) to contain clusters of T-cell epitopes. In these clusters, RASs were co-localized with the epitopes of CD8+ T cells and were predicted to modulate the immune recognition of respective peptides. These findings point that in treatment naïve CHC patients RAS may evolve due to immune pressure, and some may present examples of successful immune escape with implications to treatment response and development of CHC complications, including hepatocellular carcinoma (HCC) and extrahepatic malignancies. 

## 2. Results

### 2.1. Prevalence of RASs

A list of clinically relevant NS5A RASs was assembled based on the latest recommendations of the European Association for the Study of the Liver (EASL) [13] and review of the published data [3,4,5,7,9,18,19]. The proportion between HCV sequences for genotypes 1а, 1b, and 3а in the studied samples was 9.4% (19/202), 46.0% (93/202) и 44.6% (90/202), respectively, which represents the distribution of these genotypes on the territory of the Russian Federation. HCV sequences included in the study were deposited in GenBank under accession numbers MG953810–MG953828 (genotype 1a), MH373257-MH373349 (genotype 1b), and MH432554-MH432641 (genotype 3a). All polymorphisms detected in the amino acid positions associated with resistance to DAA are presented in Figure 1. Appendix A shows the prevalence of clinically relevant polymorphisms in the Russian NS5A sequences grouped according to HCV genotype. In the same positions, we detected also a number of amino acid polymorphisms with unknown clinical relevance (Figure 1). 

The total proportion of strains carrying RAS in NS5A was 35.1% (71/202), including 57.9% (11/19) in HCV genotype 1а, 22.6% (21/93) in HCV genotype 1b, and 43.3% (39/90) in HCV genotype 3а. HCV 1a strains harbored only one polymorphism, namely M28V, associated with significant resistance to ombitasvir [7]. Its frequency was unexpectedly high, while Q30R, L31M and Y93H substitutions associated with resistance to almost all NS5A inhibitors (daclatasvir, ledipasvir, ombitasvir, velpatasvir, and elbasvir) were not detected (Figure 1 and Appendix A). 

L31M and Y93H substitutions in HCV 1b, associated with resistance to the majority of NS5A inhibitors [7], were found with frequency 5.4% each. We detected also less clinically significant substitutions, L28M, R30Q, P58S, A92T, associated with resistance to daclatasvir (Figure 1 and Appendix A). 

A30S, associated with resistance to daclatasvir and velpatasvir and specific to HCV3a strains, was found in 31.0% of the sequences. Another variant at this position, А30K, was found in 5.7% of HCV 3a sequences. Y93H polymorphism, clinically relevant for various HCV genotypes, was found in 2.2% of HCV 3a sequences (Figure 1 and Appendix A). 

Out of 71 sequences harboring RAS, 66 (93%) contained one, and 5 (7%), two RASs: R30Q+Y93H, or P58T+Y93H, or A92T+Y93H for HCV 1b; and A30K+S62L or S62L+Y93H for HCV 3a strains (GenBank accession numbers MH373296, MH373314, MH373334, MH432639, and MH432554, respectively). 

We performed a phylogenetic analysis to see if sequences included in our study were genetically related. The analysis demonstrated that 17/19 HCV 1a NS5A sequences formed a single monophyletic group closest to HCV 1a strains originating from USA (posterior probability = 0.90), indicating that these strains originate from a single introduction of HCV 1a to Russia (Figure 2). 

These strains were found both in intravenous drug users (IDUs) and non-IDUs (Figure 2), i.e., they evenly circulated in the population. The age of this clade was calculated to be 38.9 years (95% HPD: 31.8–52.5 years). This clustering of Russian HCV 1a sequences was confirmed by the analysis of sequences of HCV genome region encoding the nucleocapsid (core) protein (Appendix A). Sequences derived from HCV genotype 1b or HCV genotype 3a were unrelated (Appendix A). However, 22 out of 26 HCV 3a sequences containing A30S formed a separate cluster, indicating that these strains might also originate from a single introduction or spread from a single de novo generated strain (Appendix A).

### 2.2. Probability of RAS Generation/Occurrence

The probability of the generation/occurrence of RAS is in part determined by the nature of the nucleotide substitution (transition or transversion) which could be represented by scores. The higher the score is calculated on the basis of the type and number of nucleotide substitutions leading to an amino acid substitution, the lower is the probability of its occurrence (which corresponds to a higher genetic barrier to acquire given polymorphism associated with drug resistance) [21]. Analysis of the nature of nucleotide substitutions leading to emergence of RAS indicates that the majority (60%–100%) result from single or dual transitions, for example, M28V in HCV 1a, or Y93H in HCV 3a sequences (Table 1). At the same time, a number of RAS has a very high genetic barrier for occurrence, such as L28M, L31M, P58T in HCV 1b, or A30K or A30S in HCV 3a sequences (Table 1).

### 2.3. Covariance of RAS with Other Amino Acid Residues within NS5A

We have used covariance analysis to assess if the observed RASs were associated with the occurrence of specific amino acid residues in other positions of NS5A linked or not to drug resistance. Surprisingly, both HCV 1b and HCV 3a sequences bearing RAS exhibited completely different covariance networks than those formed by respective sequences without RAS (except for the co-varying pair 53-54 present in the networks formed both by HCV 3a with RAS and by HCV 3a without RAS) (Figure 3a vs. 3b; Figure 3c vs. 3d). The number of amino acid residues involved in covariance interactions tended to be higher in the networks formed by sequences with RAS than those without RAS: 7 vs. 4 in HCV 1b, and 11 vs. 7 in HCV 3a (not statistically significant).

Covariance networks formed by the Russian HCV sequences involved RAS in position 28 in HCV 1a, 30 in HCV 1b and 30 and 62 in HCV 3a (Figure 3b,d,e; Table 2). Interestingly, these RASs co-varied with a set of amino acid positions not involved in the resistance (Figure 3b for HCV 1b; and Figure 3d for HCV 3a). Furthermore, an amino acid residue involved in both RAS and wild-type sequence networks, formed in these networks co-variant pairs with different amino acid residues (Table 2).

The number of Russian HCV 1a sequences was too small to run separate analysis for RAS-containing and wild-type sequences; all 19 sequences were analyzed together. Clinically relevant amino acid residue in position 28 was found to form covariance network with amino acid residues in positions 78, 308, and 372 (Figure 3e). To check if these effects are valid for a different set of sequences, we have built a covariance network for sequences from the case of HCV 1a introduction into Japan [20] (marked in green on phylogenetic tree in Figure 2). This Japanese sequence set was rich in M28V RAS (19.2%; 5/26), although not as rich as the Russian set (57,2%, 11/19; Figure 1). The covariance network for the Japanese set included amino acid position 28 as for the Russian, however, other covariant amino acid positions were different (compare Figure 3e,f).

### 2.4. Prediction of the Effect of RAS on Immune Recognition of Epitopes

We performed in silico prediction of the likelihood of occurrence of CD8+ and CD4+ T cell epitopes in all HCV 1a, 1b, and 3a wild-type and RAS-containing sequences obtained in this study and sequences from DAA-treatment naïve patients all over the world, which we retrieved from Gene Bank using pattern search for RAS in NS5A (Appendix A). The analysis was based on the set of globally prevalent HLA-A, HLA-B, HLA-C, and HLA-DR molecules and used engines of NetMHCpan-4.0 and NetMHCIIpan-3.2 servers. Epitopic profiles of CD8+ and CD4+ T cells for NS5A of HCV genotypes 1a, 1b and 3a predicted the region between aa 24 and 93, harboring the majority of known RAS, to contain a cluster of CD8+ and CD4+ T cell epitopes (Figure 4a,c,e,g). We looked for experimental proof of the enhanced immunogenicity of the RAS-reach NS5A region aa 24-93 in the IEDB database. Published data confirmed that the region between amino acid residues 28 to 32 of NS5A harbors a cluster of human CD8+ epitopes (Appendix A). Several CD8+ T-cell epitopes were also identified in the regions harboring RAS at aa 58–64, and 92–93 (Appendix A), confirming the validity of in silico scoring.

Further we calculated difference in the scores predicting CD8+ and CD4+ epitopes in the wild-type and in the RAS-containing Russian (Figure 4b,f) and “global” (Figure 4d,h) NS5A sequences. Introduction of RAS did not lead to the emergence of new epitopes (either CD8+, or CD4+) in the loci predicted to be non-immunogenic for the wild-type NS5A variants (Figure 4 b,d,f,h). In the Russian HCV 1b sequences, RAS mutations in the aa 28–32 region led to an increase in the CD8 epitopic scores (Figure 4b). However, for the “global” HCV 1b sequence set, introduction of RAS led to a considerable reduction in CD8+ T cell binding score of RAS-harboring peptides (i.e., weaker binding of respective peptides to the receptors of T cells restricted to the globally prevalent haplotypes HLA-A, HLA-B, HLA-C, and HLA-DR, resulting in a weaker anti-peptide immune response) (Figure 4d). Furthermore, RAS mutations led to a reduction in the CD8+ epitopic score for the region aa 28 to 32 of NS5A of both Russian and “global” HCV 1a and 3a sequence sets (Figure 4b,d). For the “global” set, we also observed a reduction in the predicted peptide binding by CD4+ T cells (Figure 4h).

## 3. Discussion

The primary purpose of this study was to determine the prevalence of RASs in the NS5A among treatment-naïve patients before the widespread introduction of DAAs. This data forms an important starting point for monitoring of the spread of these polymorphisms after systematic introduction of DAAs into clinical practice. 

A significant number of NS5A RASs have been recorded [3,7,8,9]. The most clinically significant are substitutions in positions M28, Q30, L31, and Y93 for HCV genotype 1a, L31 and Y93 for HCV genotype 1b, and A30 and Y93 for HCV genotype 3а. These natural polymorphisms are associated with a medium (10–100 times) or high (>100 times) increase in the half maximal effective drug concentration (EC50) in vivo and with over 10% of all antiviral treatment failures in clinical practice [7]. RASs in NS5A are of particular concern due to their potential to accumulate in the host population [22,23,24]. It has been demonstrated that NS5A variants emerging during antiviral therapy do not disappear after termination of treatment but continue to persist as quasispecies [22,23,24]. Furthermore, HCV with certain RAS (such as Y93) are associated with high incidence of developing HCC even after attaining sustained virological response [25]. This makes NS5A polymorphisms fundamentally different from the resistant variants in other DAA target proteins, in particular NS3 protease, where polymorphisms disappear quickly after the discontinuation of a failed treatment [23,26], and closer to polymorphisms in HCV core which were repeatedly shown to associate with the development of HCC [27,28]. This motivated our choice of RASs in NS5A as primary targets of the analysis of resistance to DAA in untreated patients with chronic hepatitis C in the territory naïve to DAA treatment. 

Population sequencing or a 15% cutoff for NGS sequencing is recommended by the EASL to diagnose HCV drug resistance [13]. Clinical significance of minor viral variants accounting for less than 15% of the HCV quasispecies has not been established [13]. Based on these data, the incidence of resistance-associated polymorphisms was assessed by Sanger sequencing which identifies variants present > 15%–20% of the viral population [29]. Analysis in this study was limited to HCV sequences originating from ethnically homogenous cohort collected in Moscow. Moscow is a large city with a high proportion of inhabitants who originate from diverse geographical regions of the former Soviet Union. A relevant study on the genetic variety of HCV conducted in several large Chinese cities showed that megalopolis can serve as accumulation points for various HCV subtypes from other regions [30]. Indeed, the proportion of HCV genotypes in the sequences included in this study represented the distribution of HCV genotypes across Russia [12]. With this, we expected the character and prevalence of RASs observed in Moscow HCV sequences to reflect the actual landscape across Russia before the introduction of DAAs. 

The most significant polymorphisms in NS5A at positions 30, 31, and 93 were relatively uncommon in our selection of the Russian HCV strains. The proportion of these variants in the treatment-naïve Russian patients infected with the HCV genotypes 1b and 3a (predominant in Russia [12]) was < 5.5%, except for A30S clinical significance of which is limited. Low prevalence of these RASs in the Russian sequence set mirrored their occurrence in treatment naïve patients in the North America, Europe and China [9,31,32]. Thus, while prevalence of RAS in some positions varies geographically [22,31,32], prevalence of polymorphisms in positions 30, 31, and 93 is not subjected to significant geographical variations. At the same time, the M28V polymorphism in the Russian strains of HCV genotype 1a was found to be unexpectedly frequent (57.9%). This particular RAS is associated with resistance of HCV 1a to ombitasvir and, to lesser extent, to ledipasvir, velpatasvir, and pibrentasvir [7,19,33,34]. The usual incidence of M28V polymorphism in treatment-naïve HCV 1a patients is 4–8% [8,9]. Clinically significant HCV 1a polymorphisms in NS5a positions 30, 31 and 93, as well as variant M28T associated with resistance to daclatasvir, were not detected. This seemingly discordant resistance profile observed in the Russian HCV 1a sequences prompted us to perform the phylogenetic analysis to see if these sequences were related. Indeed, 17/19 sequences were found to originate from a single introduction of HCV 1a to Russia. Thus, the high prevalence of certain NS5A RASs in Russian HCV 1a sequences and the absence of the others were due to the founder effect, that is, the strain carrying these substitutions upon its introduction to Russia served as the ancestor to most of the Russian strains of this genotype. The phylogenetic analysis showed that the monophyletic group comprising Russian HCV 1a strains is closely linked to HCV 1a sequences from the USA (where this specific polymorphism is prevalent) [22]. It should be noted that genotype 1a, highly prevalent in Europe and North America [35], is relatively uncommon in Russia, accounting for only 7% of the total population, and 9% among injection drug users [36], and hence should not pose a major threat to the successful massive use of NS5A inhibitors. 

Besides polymorphisms with known clinical significance we identified a number of novel substitutions in amino acid positions, associated with resistance such as 62 in HCV 1b, and 30 and 62 in HCV 3a. Clinical significance of these mutations is not characterized. A recent study by Carrasco et al demonstrated that changes at position 30 were more frequent in failures than cures (22.2% vs 6.4%, *p* = 0.074) [37]. Substitutions in position 62 do not contribute to baseline resistance to NS5A inhibitors, however, a linked variant Q30R-E62D was shown to confer a high-level resistance *in vitro* and was likely responsible for a viral breakthrough in vivo [38].

In total, 35% of NS5A sequences contained substitutions associated with resistance to DAA, 57.9% in HCV genotype 1а, 22.6% in HCV genotype 1b, and 43.3% in HCV genotype 3а. Out of 71 sequences harboring RAS, 66 (93%) contained one, and 5 (7%), two resistance associated substitutions. Implications of high prevalence of NS5A RASs for treatment of the Russian HCV infected patients with NS5A-targeted DAA are unclear, since presence of RASs does not necessarily lead to the treatment failure [7]. These variants may disappear from viral quasispecies over the time. Besides, HCV NS5A inhibitors are typically used in combination with NS3 or NS5B inhibitors, which ensure treatment response even in the presence of the baseline NS5A RAS. 

Testing for NS5A RASs prior to treatment is currently recommended only for: (i) patients with HCV 1a (regardless of their treatment history and stage of fibrosis) when prescribed elbasvir/grazoprevir; (ii) HCV 1a infected treatment-experienced patients when prescribed ledipasvir/sofosbuvir; and (iii) genotype 3 infected treatment-experienced patients (also treatment-naïve patients with cirrhosis) when prescribed sofosbuvir/velpatasvir or daclatasvir/sofosbuvir [39]. On one hand, our data suggest no need for special baseline resistance testing for CHC patients, except for patients infected with HCV 1a which would most probably carry M28V virus. On the other hand, it suggests that some variants (M28V in HCV 1a or A30S in HCV 3a) spread very efficiently in the host population. Hence, regular monitoring would be needed to timely detect if they increase in prevalence after the start of widespread use of NS5A inhibitors. 

Further, we attempted to analyze the factors influencing the spectrum and prevalence of the observed RAS. Ours as well as earlier published data revealed that RASs rarely occur in structurally/functionally critical amino acid positions. HCV NS5A is composed of three domains (DI, DII, and DIII) separated by two linker regions [40]. The majority of the known RAS-associated residues of NS5A protein are located within the linker between amphipathic α-helix and domain I (residues 26–32) or inside domain I (residues 33–213) not involved in the protein–protein interactions or protein phosphorylation [14]. DI domain, and full-length NS5A were shown to form dimers [41,42] suggested to serve as a scaffold for viral replication [43]. Despite localization in DI, none of RAS affect dimerization of NS5A [15,41,44]. Also, RAS do not affect most of the NS5A activities and do not intervene into the respective amino acid motives (see D. Ross-Triepland and M. Harris for review [14]). Namely, they do not affect the positions known to be involved in maintaining of NS5A structure and activities, such as: AH anchoring of NS5A to ER (aa 5-25); Zn^2+^ binding motif (Cys-39, Cys-57, Cys-59, and Cys-80); the lipid droplet-binding motif (aa 100-104); PI4KIIIα-binding motif (aa 202-210); sites of phosphorylation (S235, and also S222 and S238); CypA-binding site (aa 311–318); P2 polyproline SH3-binding motif (aa 343–356); or highly conserved basic cluster at the N-terminus of DIII, critical for particle assembly (aa 352–355) [14]. Super-resolution microscopy demonstrated that although HCV treatment with DAA led to inhibition of RNA replication and reduction in NS5A cluster size, this phenotype is maintained also in the presence of the Y93H resistance associated substitution, i.e., Y93H do not causedetectable changes on the structure of NS5A replication clusters [45]. An exception are substitutions in the CypA-binding site in aa position 318 which confer resistance to samatasvir [46]. Due to this “uninvolvement”, RASs in NS5A have lower impact on the viral fitness than mutations in HCV protease or polymerase.

Although the majority of RASs is not directly involved in NS5A functions, they may act by inferring certain structural changes allowing viral escape at a cost affordable for virus replication. This could be revealed by the analysis of protein covariance networks. Earlier studies demonstrated an absence of the covariance of residues in the resistance associated amino acid positions in NS5A with residues in other HCV proteins [47]. Only one of the clinically significant polymorphisms in the NS5A HCV protein, namely R30, was found to co-vary with amino acid positions in other HCV proteins [48]. We have re-addressed this issue and performed a covariance analysis for the full-length NS5A sequences with and without RAS which gave an opposite result. Most of NS5A covariance networks we observed included at least one amino acid residue involved in resistance to DAA (such as aa 28, 30, 62). Furthermore, in HCV 1b and 3a sequence sets presence of RASs was found to lead to drastic changes in the NS5A covariance patterns. Last but not the least, amino acid residues in position 28 of HCV 1a, and in position 30 of HCV 1b and HCV 3a in RAS and wild-type covariance networks were associated with different amino acid residues. Repeating the scenario for RAS, amino acid residues co-varying with RAS were not localized in the regions of NS5A involved in drug resistance or maintenance of NS5A structure and functions (with the exception of aa residues in lipid droplet-binding motif at aa position 103 and CypA-binding site at aa 313, 315). Biological meaning/implications of these coordinated amino acid changes are unclear. Phylogenetic analysis done by Knops with coauthors revealed significant association of R30Q in HCV 1b with secondary mutations Q24K/R and V34L/I and, less significant, with V138L and L183P (aa position 138 is involved our covariance network for HCV 1b; Figure 3b). Authors attributed these associations to undefined epistatic interactions [49]. Altogether, these data support the concept of drug resistant HCV variants as a natural part of viral population evolving through complex coordinated changes in NS5A occurring with minimal involvement of NS5A structure and functions. In other words, HCV strains resistant to DAA evolve and expand/spread thanks to high evolutional freedom within NS5A, with minimal effects on the replication fitness of the virus [50]. Another proof of RAS resulting from the coordinated changes within NS5A not linked to application of DAA and to the strive for viral fitness could be found in the analysis of the likelihood of their de novo emergence. The latter is largely dependent on the mutation type: whether they are transitions or transversions. Using a model where the probability of transitions in the HIV genome is 2.5 times higher than that of transversions [51], Kliemann et al. developed a system to rank the probability of overcoming the genetic barrier to drug resistance for HCV [21]. Using this system, we found that although many RASs occur through favorable transitions, certain RASs of clinical significance, such as L28M, L31M, P58T in HCV 1b, and A30K or A30S in HCV 3a sequences, result from the less favorable transversions, or even combinations of transversions and transitions with a genetic barrier over 2.5 (reaching 5 to 6 for A30K and A30S). For example, A30K/S in HCV 3a due to transversions was highly prevalent, whereas Y93H due to transition was quite rare (5.5% in genotype 1b compared to the 10.6% found in the literature [9]). Prevalence of the “high cost” substitutions indicates that these RASs offer to NS5A complex advantages other than simple biochemical fitness.

Minimal effects of RAS for the virus (in the absence of DAA) raise a question of the driving force of these coordinated changes, type of the evolutional pressure, and the nature of losses and/or benefits outside of NS5A structure and functions. We attempted to delineate the mechanism(s) behind the appearance of these substitutions. The major force driving viral evolution is adaptation to the host. Mutations correlating with response to the IFN treatment revealed geographic variation explained by differential impact of the host-related factors [52]. We have attributed spread in Russia of HCV 1b strains with mutations in the core protein, associated with resistance to IFN treatment such as R70H/Q, to an escape from the immune pressure by HLA types prevalent in this territory [53]. Bioinformatic study by Cuypers L. et al. indicated that NS5A harbors clusters of B- and T cell epitopes overlapping regions harboring RAS [54]. In a later study Ikram A. et al. explored polymorphic regions of HCV, including RAS in NS3, NS5A, and NS5B, for the presence of CD4+ and CD8+ T cell epitopes, and found several hot spots in which immune and drug selective pressures overlapped [55]. According to Cuypers L. et al., RASs at aa 28 to 32 of NS5A were co-localized with CD4+ T- and B-cell epitopes, and at aa 58–62, with B-cell epitopes (but not with CTL epitopes), whereas conservation showed negative correlation to the presence of T-cell epitopes (immunological constraints), specifically significant for NS5A, also in the regions harboring RAS [54]. Here, we have on contrary shown that RASs in NS5A region aa 26 to 32 co-localize with a cluster of epitopes of CTL T-cells. Several T-cell epitopes were also found in the regions harboring aa 58–64, and 92–93 (Appendix A). Furthermore, our in silico analysis showed that introduction of RAS caused changes of the epitopic scores of respective regions. A decrease was observed of CD8+ epitopic scores for the region aa 28 to 32 of NS5A for both Russian and “global” HCV 1a and 3a sequence sets. For the “global” set, we also observed a reduction in the predicted peptide binding by CD4+ T cells. RAS in the region between aa 28-32 of Russian HCV 1b led to an increase in peptide binding scores. However, for the global HCV 1b sequence set, introduction of RAS in this region led to a decrease in the peptide binding scores for both CD8+ and CD4+ T cells. Causes of the differential effect of RAS on the epitopic scores for the Russian and global HCV 1b sequences may relate to prevalence of certain HLA types in these subgroup of patients, and need to be further elucidated, but on the overall, introduction of RAS in the region aa 28-32 of NS5A led to a decrease in epitopic/peptide binding scores.

Decreased epitopic score predicts a loss (partial loss) of immune recognition of the RAS-harboring region. These in silico results have experimental proof. Introduction of RASs in NS3 (Q80, V55I) and NS5B (M423I, V499A) completely prevented recognition of the epitopes associated with HCV clearance by T-cells specific to the wild-type sequences [55,56]. In this context, the appearance of certain RASs within NS5A in treatment naïve patients could reflect an outcome of the successful immune escape contributing to the preferential spread of respective viral variants. This could be the case for HCV 3a with A30S or HCV 1a with M28V upon its introduction in the Russian population. Unique proof of such HLA/immune response directing viral evolution was obtained in a study of individuals with chronic HCV infection infected from a single HCV genotype 1b source [57]. This data supports our concept of immune pressure driving divergent asymmetrical evolution of HCV 1a after single introduction into Russia and into Japan reflected by difference in their covariance networks (Figure 3e,f). Interestingly, HCV 1b sequences included in the current study revealed increase in the prevalence of HCV 1b strains with the immune escape substitution M91L in the core protein. Furthermore, 3 out of 26 (11.5%) of viral isolates bearing substitutions in aa positions 70 and/or 90 of HCV core contained also RAS in aa positions 30 or 31 of NS5A indicating dual immune escape (GenBank accession numbers MN026555, MN026562, MN026684 for core, and MH373284, MH373296, MH373331 for NS5A sequences, respectively) indicating coordinated viral evolution towards weaker recognition by human immune system. Altogether, this stresses the importance of the host factors as an immunogenetic background that shapes the immune pressure on the virus, and hence, its variability. To conclude, the resistance to NS5A-targeting DAA is a complex phenomenon in which adaptation of virus to replication in the presence of drug may be reinforced by the capacity of this “adapted” virus to escape host immune response, persist and trigger HCV-associated malignancies. 

In conclusion, we found that spectrum and prevalence of HCV NS5A RASs are influenced by several factors, such as: i) genetic barrier to generate RAS; ii) T cell immune pressure and primary immune escape; iii) further functional and/or immunological adaptation reflected by amino acid covariance; and iv) founder effect in some of HCV genotypes/strains. The next step will be to assess the prevalence and circulation of RAS-containing HCV strains in different ethnical groups several years after the introduction of DAAs into the clinical practice. Such studies would reveal if certain NS5A RASs can give HCV an advantage in terms of its preservation and spread in the host population in the context of the broadening use of antiviral treatments.

## 4. Materials and Methods 

### 4.1. Human Serum Samples 

HCV RNA sequences were obtained from 202 samples of patient sera collected between 2008 and 2014 in Moscow from high-risk individuals (injection drug users, *n* = 67) and low-risk individuals (*n* = 135). Serum samples collected in 2008–2011 came from patients with clinically diagnosed chronic hepatitis C followed at the Hepatology Center of Infectious Disease Hospital No. 1 (Moscow, Russia). Serum samples collected in 2014 were obtained from anti-HCV positive patients followed at the toxicological emergency unit of the Sklifosovsky Research Institute of Emergency Medicine (Moscow, Russia). All patients were of the Caucasian ethnicity and were residents of Moscow city. The study was conducted according to the principles expressed in the Declaration of Helsinki. Written informed consent to take part in the study was obtained from all participants. The study design was approved by the Ethics Committee of the Chumakov Federal Scientific Center for Research and Development of Immune-and-Biological Products of Russian Academy of Sciences, Moscow, Russia (Approval #10 dated 23 May 2016).

### 4.2. RNA Isolation and RT-PCR 

RNA was isolated from blood samples using the QIAamp Viral RNA Mini Kit (QIAGEN, Hilden, Germany), MagNA Pure Compact Nucleic Acid Isolation Kit I (Roche Applied Science, Mannheim, Germany) and Sileks MagNA (Sileks, Moscow, Russia), following the protocols of their respective manufacturers. HCV RNA detection was carried out by RT-PCR using primers specific to the most conserved region of the virus genome, the 5′ untranslated region (5′ UTR). The following primers were used to amplify 5′ UTR sequences: external forward 5′-ctg-tga-gga-act-act-gtc-tt-3′, external reverse 5′-tat-cag-gca-gta-cca-caa-gg-3′; internal forward 5′-ttc-acg-cag-aaa-gcg-tct-ag-3′; and internal reverse 5′-acc-caa-cac-tac-tcg-gct-ag-3′. The following conditions were used for the first round of PCR: reverse transcription at 42 °С for 60 min, 94 °С for 5 min, then 35 cycles of denaturation at 94 °С for 30 s, annealing at 55 °С for 30 s and elongation at 72 °C for 45 s, and a final elongation period at 72 °С for 7 min. The product of the first PCR was amplified in a second round of PCR with the following conditions: 94 °С for 5 min, followed by a 15 cycles of denaturation at 94 °С for 30 s, annealing at 55 °С for 30 s and elongation at 72 °C for 45 s, and the final elongation period at 72 °С for 7 min. The length of the resulting fragment was 207 nt.

### 4.3. HCV Genotyping 

The genotype of the virus was determined in all samples positive for HCV RNA. For the samples collected during the years 2008–2011, the genotype was determined RT-PCR with the type-specific primers suggested by Ohno et al. [58]. For the samples collected after 2011, HCV genotyping was performed by analyzing amplified nucleotide sequences within the core protein-coding fragment of the virus genome. HCV core amplification was performed as described elsewhere [53]. HCV core sequencing data are deposited in GenBank under accession numbers MN026550-MN026739.

### 4.4. Amplification and Sequencing of HCV NS5A Coding Region 

Genotype-specific primers were used to amplify the HCV NS5А fragment. For genotype 1, the following primers were used: external forward 5′-arg-agr-ctn-cay-car-tgg-at-3′, external reverse 5′-crc-chg-tcc-ang-wrt-arg-ac-3′, internal forward 5′-gay-rty-tgg-gac-tgg-ath-tg-3′, and internal reverse 5′-ctc-acv-gtn-gac-cad-gac-c-3′. For genotype 3, the following primers were used: external forward 5′-art-gga-tya-atg-arg-ayt-ayc-3′, external reverse 5′-rcc-rgt-cca-rga-rta-yga-c-3′, internal forward 5′-cat-ctg-gga-htg-ggt-htg-3′, and internal reverse 5′-gac-car-gar-tcr-car-ctc-aa-3′. Reverse transcription and amplification were performed using the Transcriptor First Strand cDNA Synthesis Kit and Fast Start High Fidelity PCR System (Roche Applied Science, Mannheim, Germany), according to the manufacturers’ protocols. The following conditions were used for the first round of PCR: 94 °С for 2 min; then 35 cycles of denaturation at 94 °С for 30 s, annealing at 56 °С for genotype 1 and 52 °С for genotype 3 for 30 s and elongation at 72 °C for 2 min, and a final elongation period at 72 °С for 7 min. The product of the first PCR was amplified in the second phase of PCR under the same conditions, except for the annealing temperature, which was 50 °С for genotype 1 and 52 °С for genotype 3. The length of the resulting fragment was 1292 nt for genotype 1 and 1296 nt for genotype 3.

All the products were extracted from the agarose gel using the QIAquick Gel Extraction Kit (QIAGEN, Hilden, Germany). The primary nucleotide sequence was determined using the 3500 Genetic Analyzer (ABI, Foster City, CA, USA) and BigDye Terminator v 3.1 Cycle Sequencing Kit. The HCV genotype 1a H77 reference strain was used to align the sequences and determine the order of the amino acids (GenBank AF011753). HCV NS5A sequencing data are deposited in GenBank under accession numbers MG953810–MG953828, MH373257-MH373349, and MH432554-MH432641. Alignment of the nucleotide and predicted amino acid sequences of HCV were performed using MEGA 7.0.18. The frequency of clinically significant polymorphisms in NS5A amino acid positions 28, 30, 31, 54, 58, 62, 92, and 93 was calculated using Microsoft Office Excel. The data were analyzed using Graphpad.com. Statistical significance was evaluated using an F-test, with *p* < 0.05 considered statistically significant.

### 4.5. Retrieval of the Wild-Type and RAS Carrying HCV NS5A Sequences from Gene Bank

For comparative in silico sequence analysis, additional HCV NS5a sequences from DAA-treatment naïve patients from all over the world were retrieved from Gene Bank. For this, 8- to 10-mer peptide patterns representing regions of NS5A harboring RAS, representing the wild-type and RAS sequences, were generated using pattern syntax exploited by www.prosite.expasy.org. Patterns were used to search translated NS5A nucleotide sequences of the full-length NS5A HCV 1a, 1b and 3a of treatment naïve chronic HCV patients. Retrieved sequences were grouped into sets according to the genotype and presence or absence of RAS: wild-type HCV 1a (*n* = 50), 1b (*n* = 50), 3a (*n* = 50), and RAS-containing HCV 1a (*n* = 30), 1b (*n* = 40), 3a (*n* = 14). The complete list of sequences is presented in Appendix A. 

### 4.6. Phylogenetic Analysis 

Phylogenetic trees for HCV sequences were built using PhyML 3.0 under a GTR model (http://www.atgc-montpellier.fr/phyml/), with SPR tree improvement (http://www.atgc-montpellier.fr/download/papers/phyml_spr_2005.pdf), and aLRT SH-like test [59]. Additionally, time-scaled phylogenetic analysis was performed for HCV 1a sequences. Temporal signature was validated with TempEst v1.5 (previously known as Path-O-Gen) (omicX, Le-Petit-Quevilly, France) by testing for a statistically significant linear correlation between the root-to-tip distance and isolation date of each sequence on a maximum likelihood phylogeny. The analysis was done using a Bayesian likelihood-based algorithm implemented in BEASTv1.8.4 [60]. The SRD06 nucleotide substitution model [61] was used with a constant population size model and a strict molecular clock model. These parameters were chosen as the best indicators after model comparison. The MCMC (Markov chain Monte Carlo) were run for 100 million generations and sampled every 10,000 steps. Tracer v1.6 (available at: http://beast.bio.ed.ac.uk/) was used for check of convergence. Effective sample size >1000. Trees were annotated with TreeAnnotator v.1.8.4 using a burn-in of 1000 trees and visualized with FigTree v.1.4.3. After running, the clock rate was 1.127 × 10^−3^ substitutions/site/year (95% higher posterior density (HPD) 8.5 × 10^−4^–1.4 × 10^−3)^.

### 4.7. Analysis of Amino Acid Covariance 

Covariance analysis for translated NS5A sequences was performed using Fastcov algorithm with score of 0.7 used as a cutoff for selecting covariant aa pairs [62]. Covariance analysis was run separately for following five batches of sequences included in this study: genotype 1a (19 sequences), genotype 1b without RAS (72 sequences), genotype 1b with one or more identified RAS (21 sequences), genotype 3a without RAS (51 sequences), and genotype 3a with one or more identified RAS (39 sequences). Additional covariance analysis was run for HCV 1a sequences set from Japan (*n* = 26). Amino-acid covariant networks were visualized using the R package igraph [63].

### 4.8. In Silico T-Cell Epitopic Analysis 

Analysis of occurrence of T cell epitopes within NS5A of HCV of diverse genotypes was done using on-line bioinformatics tools available at Immune Epitope Database (IEDB, https://www.iedb.org/). Conservation of these epitopes was found by aligning selected NS5A sequences using MultAlin online software and conservancy analysis tool available at IEDB.

To access the effect of RAS on immune recognition of epitopes, we predicted peptide binding to MHC class I molecules using NetMHCpan-4.0 server at www.cbs.dtu.dk/services/NetMHCpan/, and to MHC class II molecules using NetMHCIIpan-3.2 server at www.cbs.dtu.dk/services/NetMHCIIpan/. We predicted CD8 and CD4 epitopes respectively in all HCV 1a, 1b, and 3a wild-type and RAS containing sequences for a set of prevalent HLA-A, HLA-B, HLA-C (40 in total) and 42 HLA class II molecule covering 31 prevalent HLA-DR (as defined from a worldwide population extracted from the Allele Frequency Net Database [64]), and also 5 HLA-DP and 6 HLA-DQ molecules. Next, wild-type and RAS epitope profiles were constructed for each HCV subtype by for each protein position calculating the number of predicted HLA binding peptides (using a prediction threshold of 0.5% rank score for class I and 10% rank threshold for class II) overlapping that position weighed by the frequency of the given HLA. This epitope profile was further normalized so that the max value for each subtype and wild type, RAS combination is equal to 1. Finally, the difference between the wild-type and RAS epitope profile for each HCV subtype was calculated. 

### 4.9. Statistical Analysis 

Data analysis was performed using the Graphpad.com. Statistical significance was evaluated by Fisher’s exact test using parametric model, two-tailed *p* value equals (*p* value < 0.05) was considered statistically significant. 

## Figures and Tables

**Figure 1 biomedicines-08-00080-f001:**
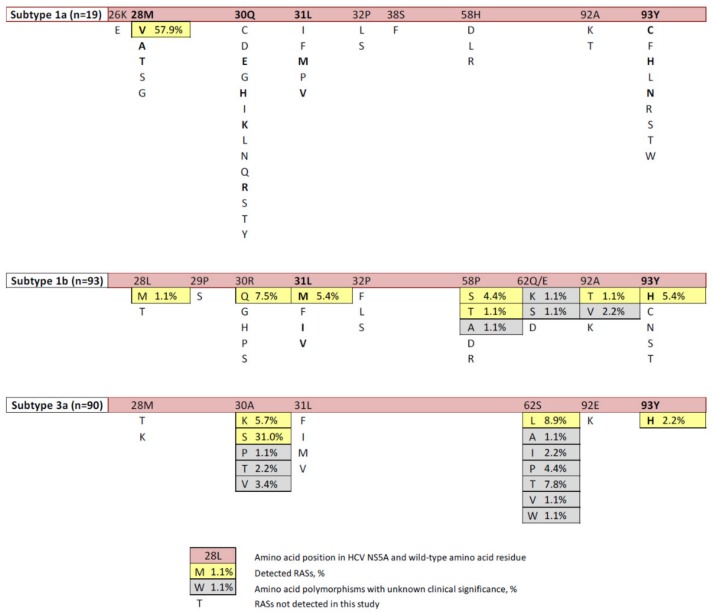
Complete list of amino acid polymorphisms in hepatitis C virus (HCV) NS5A associated with resistance to direct-acting antivirals (DAA). Resistance-associated substitutions (RASs) in bold are selected in >10% of virologic failures (based on the published data [3,4,5,7,9,13,18,19].

**Figure 2 biomedicines-08-00080-f002:**
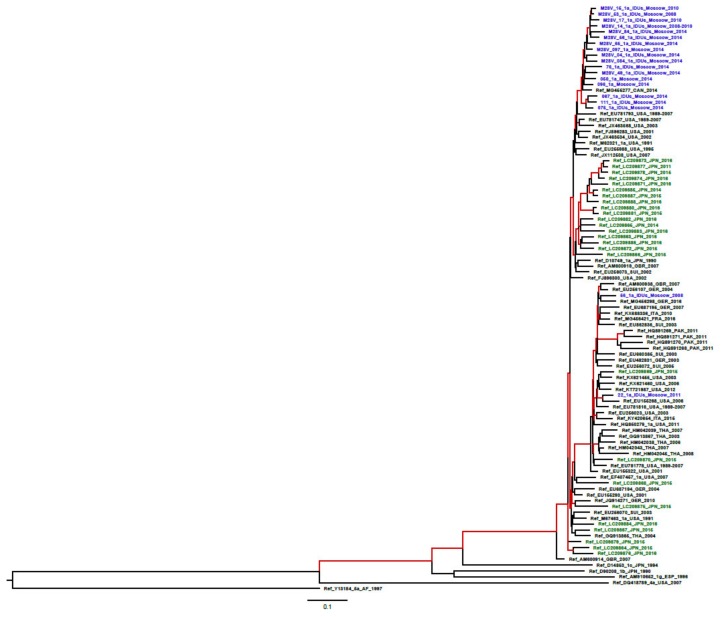
Phylogenetic tree for HCV 1a NS5A sequences (1126 nucleotides, positions 6330–7455 according to HCV 1a reference strain H77 (GenBank NC004102)), built under a GTR model; branches with group reliability >90% are indicated in red; HCV 1а sequences from this study are indicated in blue; the type of RAS (M28V) or wild-type (WT) are indicated for each sequence. HCV1a sequences from Japan [20] are indicated in green. Sequences isolated from the intravenous drug users marked as IDUs.

**Figure 3 biomedicines-08-00080-f003:**
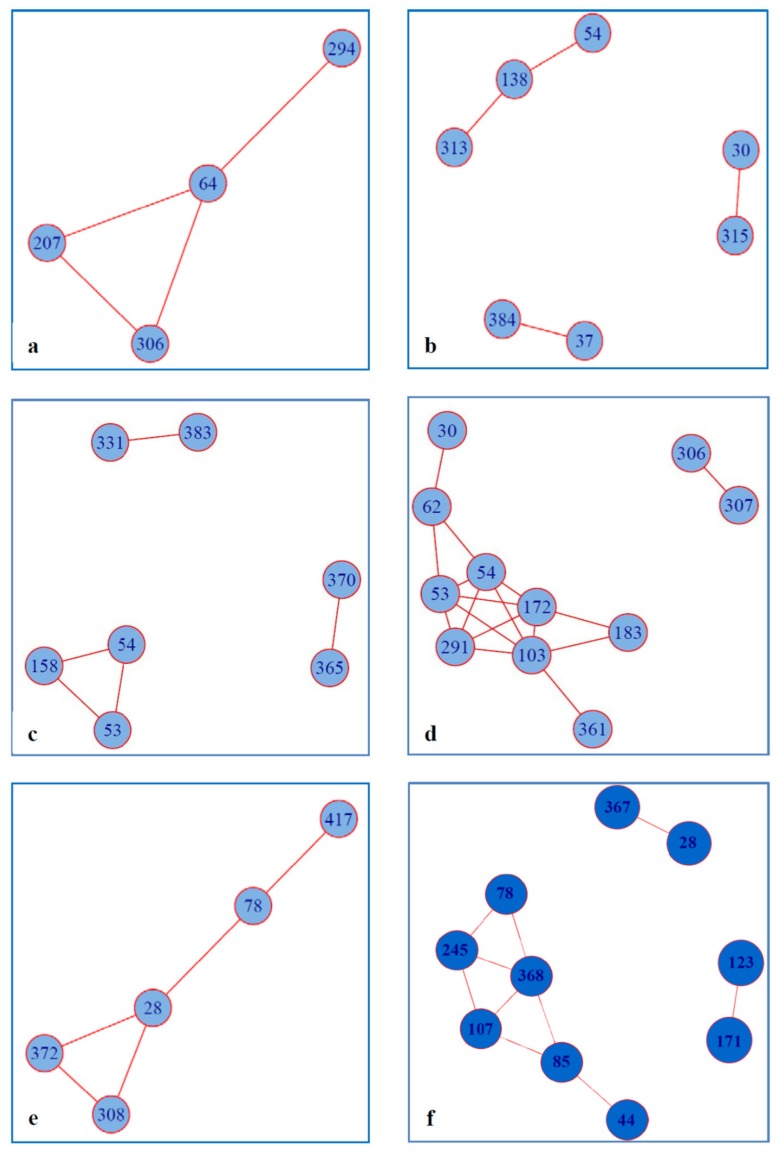
Amino acid covariance networks for NS5A sequences built for HCV 1b without RAS (*n* = 72) (**a**); HCV 1b with at least one RAS (*n* = 21) (**b**); HCV 3a without RAS (*n* = 51) (**c**); HCV 3a with at least one RAS (*n* = 39) (**d**); HCV 1a (*n* = 19) (**e**); HCV 1a from Japan (*n* = 26) (**f**). Amino acid covariances within alignments of the HCV sequences were graphed with the covarying positions (nodes) represented as circles and the covariances between the positions (edges) as lines. Covariance networks were built using algorithm named ‘Fastcov’ implemented in the golang (https://golang.org) programming language (https://www.ncbi.nlm.nih.gov/pmc/articles/PMC4958985/). The executable binary files are available at http://yanlilab.github.io/fastcov. Networks were visualized using the R package igraph.

**Figure 4 biomedicines-08-00080-f004:**
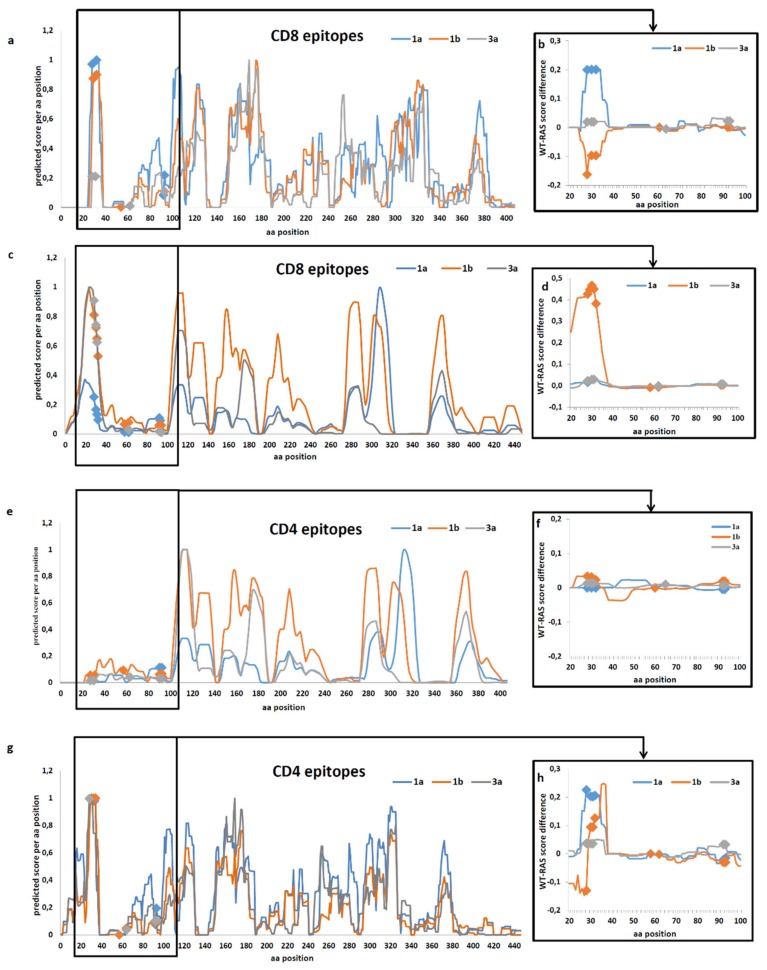
Epitopic profiles of the wild-type NS5A of HCV 1a, 1b, and 3a predicting localization of CD8+ T cell epitopes in the Russian (**a**) and “global” (**c**) sequences, and CD4+ T cell epitopes in Russian (**e**) and “global” (**g**) sequences; Difference of the epitopic scores of HCV 1a, 1b and 3a with and without RAS in RAS-harboring region between aa 20 to 100 for CD8+ T cell epitopes in Russian (**b**) and “global” (**d**) sequences, and CD4+ T cell epitopes in Russian (**f**) and “global” (**h**) sequences, positions of RAS are highlighted on curves in diamonds. Predictions were done using NetMHCpan-4.0 and NetMHCIIpan-3.2 and were based on a set of globally prevalent HLA-A, HLA-B, HLA-C and HLA-DR molecules extracted from the Allele Frequency Net Database and 5 HLA-DP and 6 HLA-DQ molecules. Epitope profiles were constructed for each HCV subtype by for each protein position calculating the number of predicted HLA binding peptides overlapping that position weighed by the frequency of the given HLA and normalized so that the max value is 1 in each run of analysis.

**Table 1 biomedicines-08-00080-t001:** Nature of detected HCV NS5A nucleotide substitutions and associated genetic barrier to resistance.

HCV Genotype (nn of Sequences)	RAS	Nn (%)	Nt Substitution						Genetic Barrier to Resistance **
			Codon, Wild Type (Prevalence, %)		Codon, RAS (Prevalence, %)		Pattern of Substitution	Nt Substitution Type *	
1а (19)	M28V	11 (57.9%)	ATG	100%	GTG	100%	A_ _→G_ _	Ts	1
1b (93)	L28M	1 (1.1%)	CTG	85.9%	ATG	100%	C_ _ →A_ _	Tv	2.5
			CTA	8.7%			C_A→A_G	Tv+ Ts	3.5
	R30Q	7 (7.5%)	CGG	68.6%	CAA	14.3%	_GG→ _ AA or _G_→ _ A_	Ts+ Ts or Ts	2 or 1
			CGA	25.6%			_G_→ _ A_ or _GA→ _ AG	Ts or Ts+ Ts	1 or 2
			CGC	2.3%	CAG	85.7%	_GC→_AA or _GC→_AG	Ts+Tv	3.5
			CGT	1.2%			_GT→_AA or _GT→_AG	Ts+Tv	3.5
	L31M	5 (5.4%)	TTG	37.5%	ATG	100%	T_ _ →A_ _	Tv	2.5
			TTA	35.2%			T_ A →A_ G	Tv+ Ts	3.5
			CTA	10.2%			C_A→A_G	Tv + Ts	3.5
			CTG	6.8%			C_ _ →A_ _	Tv	2.5
	P58S	4 (4.4%)	CCA	84.0%	TCA	100%	C_ _→T_ _	Ts	1
			CCG	9.2%			C_G→T_ A	Ts+ Ts	2
			CCC	1.1%			C_C→T_ A	Ts+Tv	3.5
	P58T	1 (1.1%)	CCA	84.0%	ACA	100%	C_ _→ A_ _	Tv	2.5
			CCG	9.2%			C_G→A_A	Tv+ Ts	3.5
			CCC	1.1%			C_C→A_A	Tv+ Tv	5
	A92T	1 (1.1%)	GCG	60.0%	ACG	100%	G_ _→A_ _	Ts	1
			GCA	32.2%			G_A→A_G	Ts+ Ts	2
			GCC	2.2%			G_C→A_G	Ts+Tv	3.5
			GCT	1.1%			G_T→A_G	Ts+Tv	3.5
	Y93H	5 (5.4%)	TAC	93.2%	CAC	100%	T_ _→C_ _	Ts	1
			TAT	4.5%			T_T →C_C	Ts+ Ts	2
3a (90)	А30К	5 (5.7%)	GCG	92.2%	AAG	100%	GC_→AA_	Ts+Tv	3.5
			GCA	5.9%			GCA→AAG	Ts+Tv+ Ts	4.5
			GCC	1.9%			GCC→AAG	Ts+Tv+Tv	6
	A30S	26 (31.0%)	GCG	92.2%	TCG	100%	G_ _→T_ _	Tv	2.5
			GCA	5.9%			G_A→T_G	Tv + Ts	3.5
			GCC	1.9%			G_C→T_G	Tv+ Tv	5
	S62L	8 (8.9%)	TCA		TTA	75%	_C_→_T_	Ts	1
				100%	TTG	12.5%	_CA→_TG	Ts+ Ts	2
					CTA	12.5%	TC_→CT_	Ts+ Ts	2
	Y93H	2 (2.2%)	TAC	100%	CAC	100%	T_ _→C_ _	Ts	1

* Transition, Ts; Transversion, Tv. ** Calculated based on the type of nucleotide substitution leading to an amino acid substitution (transition—1 point, Tv—2.5 points) [21]. Nucleotides in triplets that remain unchanged are indicated with underscore. The most significant mutations (associated with > 10% of virologic failures) are marked in bold.

**Table 2 biomedicines-08-00080-t002:** NS5A covariance pairs associated with RAS.

HCV1a		HCV1b		HCV3a	
RAS	Wild Type	RAS	Wild Type	RAS	Wild Type
28V-78R	28M-78K	30Q-315V	30R-315I	30S-62S	30A-62L
28V-308L	28M-308R				
28V-372L	28M-372V				

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
