# Peer review of "Factors Influencing the Prevalence of Resistance-Associated Substitutions in NS5A Protein in Treatment-Naive Patients with Chronic Hepatitis C"

_biomedicines, 2020, doi:10.3390/biomedicines8040080_

Round 1

Reviewer 1 Report

In this study, the authors analyzed the prevalence of resistance-associated amino acid substitutions (RASs) in hepatitis C virus (HCV) NA5A protein in the patient population in the Russian Federation. The authors found that majority of their detected RASs, potentially to be associated with resistance to direct-acting antivirals (DAAs), are within the polymorphic regions of NS5A. They also found that some specific residue substitutions, which have not previously reported to be associated with resistance to DAAs, are sometimes co-varied with these resistance-associated RASs. Finally, the authors performed some in silico analysis to predict the potential effects on these RASs on immune recognition by CD8+ and CD4+ T cells.

This manuscript is well-written and provides baseline landscape information for RASs in HCV NS5A before DAA treatment. This study is very useful for understanding the treatment outcome, the evolution of these RASs over the treatment, and has implications to understand potential disease progression in some patients. There are, however, some very minor concerns that need to be further addressed.

  1. As the authors pointed out, some studies have shown that NS5A substitutions persist as quasispecies after termination of treatment. For the HCV strains isolated from each patient, did the authors also find quasispecies for NS5A in treatment-naïve patients? This might give us some clue on the driving force for evolution of RASs during the treatment? This might also be interesting to discuss as NS5A could be trans-complimented during viral life cycle.
  2. NS5A is an indispensable component for HCV replication and also one of the main targets of current DAAs. Therefore it is totally understandable that the authors put their focus on substitutions within NS5A protein. Given that NS5A also interacts with many other HCV structural and nonstructural proteins that are also important for viral life cycle, it might be interesting to pay some attentions to the substitutions in other proteins. Some of these substitutions may compensate or compliment NS5A functions.
  3. The title says “factors influencing…”, however, the current version of this manuscript does not provide enough information about such factors. The authors did not mention in the discussion section that “the majority of the known RASs within NS5A are located within the linker…”. This domain I of NS5A has been reported to be important for RNA binding, for membrane anchor, and for zinc binding, with some key residues being reported. It might be interesting to further discuss these functions of NS5A in the context of RASs.

Author Response

1.

As the authors pointed out, some studies have shown that NS5A substitutions persist as quasispecies after termination of treatment. For the HCV strains isolated from each patient, did the authors also find quasispecies for NS5A in treatment-naïve patients?

This might give us some clue on the driving force for evolution of RASs during the treatment? This might also be interesting to discuss as NS5A could be trans-complimented during viral life cycle.

We are very grateful to Reviewer for comments and thorough analysis of our paper.

Indeed, there is a possibility that many of treatment-naïve patients have such RASs as minor variants in quasi species that may be selected during therapy. Our primary goal was to assess the prevalence of RASs that are not minor, but already make up a substantial, even major part of the quasi specie population. We used Sanger sequencing which identifies variants present > 15-20% of the viral population. We mention this in Discussion section (page 11, lines 276-280) and in Materials and Methods section (page 16, lines 503-505). The occurrence of viral variants with RAS as minor populations can be even more wide-spread. Viral variants with such mutations co-infecting cells together with non-mutated variants can possibly complement them. This is a very interesting hypothesis, supported by finding of recombinant HCVs (Kalinina O et al, 2006 and later studies).

2.

NS5A is an indispensable component for HCV replication and also one of the main targets of current DAAs. Therefore it is totally understandable that the authors put their focus on substitutions within NS5A protein. Given that NS5A also interacts with many other HCV structural and nonstructural proteins that are also important for viral life cycle, it might be interesting to pay some attentions to the substitutions in other proteins. Some of these substitutions may compensate or compliment NS5A functions.

This is a very interesting issue. Despite extensive interactions of NS5A with other HCV proteins, earlier studies demonstrated an absence of the covariance of residues in the resistance associated amino acid positions in NS5A with residues in other HCV proteins (except amino acid position 30 in HCV-1b). This was referred to as one of the reasons behind high variability within NS5A – as variations within NS5A need not be coordinated with or compensated by changes in other HCV proteins. This information available prior to the study motivated our selective sequencing of NS5A coding region. Accumulation of HCV sequence information after massive application of DAA may show that scenario for DAA resistant HCV variants is different. New prospective HCV studies are needed to address this issue.

3.

The title says “factors influencing ...”, however, the current version of this manuscript does not provide enough information about such factors.

The authors did not mention in the discussion section that “the majority of the known RASs within NS5A are located within the linker ...”. This domain I of NS5A has been reported to be important for RNA binding, for membrane anchor, and for zinc binding, with some key residues being reported. It might be interesting to further discuss these functions of NS5A in the context of RASs.

According to our data, the factors which influence the presence or absence of certain RASs are: i) genetic barrier to generate RAS; ii) T cell immune pressure and primary immune escape; iii) further functional and/or immunological adaptation reflected by amino acid covariance; iii) T cell immune pressure; iv) founder effect in some of HCV genotypes/strains. We summarized factors in conclusions (page 15, lines 444-447).   

To broaden the discussion of NS5A functions in the context of RASs, we added the list of amino acid residues that are shown to be important for NS5A structure and functions. Interestingly, none of these residues were involved in covariance patterns detected in our study (page 13, lines 341-349).

Reviewer 2 Report

In this article, Kyuregyan et al. sequenced residue substitutions in DAA treatment-näive patient cohort infected with Hepatitis C virus genotypes 1a, 1b and 3a and analyzed some key substitutions in viral NS5A protein in a great proportion of the patients. Although massive analyses were performed, the authors messed up scientific roles of these substitutions in viral replication, potential virus resistance to the DAAs and viral immune escape. According to current in silico studies, the authentic and exact role of the substitutions however is ambiguous. Furthermore, it is not proper to define them as "resistance"-associated substitutions before DAA treatment, since DAA has not been administrated yet. They are mutations during replication but not substitutions passively driven by DAA treatment, although they showed the same mutative directions in comparison to identified RASs after treatment.

Major comments:

1. RAS stands for resistance-associated substitution upon DAA treatment, which is distinct from viral mutation. RAS is a passive process, a selective effect by DAAs that kill replication of wild-type virus. However, mutation is active and spontaneous probably driven by viral replicative advantange or host immune pressure, however neither is associated with DAA treatment in treatment-näive patients. HCV could not predict a future DAA treatment and evolves RASs in advance. Also a major problem is that the word "DAAs" is present in article and abstract but only in Figure 1 and Table 1, other data are not directly related with DAA.

2. I have to argue that as a RNA virus, HCV replication creates thousands of mutations in treatment-näive patients every day -in this regard- mutations on all viral proteins may be of importance, although NS5A is one of the most essential component but not the sole in viral replication. In this case, mutations (or substitutions) are either promoting viral replication or escaping immune recognition, again if you are talking about treatment-näive patients. However after DAA treatment RASs are pretty specific, since DAAs are direct-acting on one viral protein (for instance daclatasvir is a specific NS5A inhibitor) therefore only after treatment that targets one protein without affecting others we can say that the substitutions are resistance-associated. In sum, one virotype resided in the patient reflects a balance of viral replication and immune pressure, later on resistance becomes a topic only when treatment starts.

3. What are biological functions and fates of these characterized substitutions after DAA treatment? As expected wild-type virus is sensitive to NS5A inhibitor and virus with substitutions becomes dominant. Overall effect would be that those patients are not suitable for daclatasvir. Whether they are responders to other agents in the cocktail regimen however is largely unknown. It would be better to moniter patients after DAA treatment, measure virus polymorphism and more importantly present a concrete therapeutic plan specific for those patients.

4. Among NS5A substitutions Leu31(L31) and Tyr93(Y93) are well-characterized. Kohler et al. (2014) have presented a predicted structure of NS5A carrying the mutations and NS5A inhibitor. If M28V (genotype 1a) and A30S (3a) interfere with NS5A-inhibitor binding, such in silico structure is required.

5. The authors showed that both nucleotide transition and transversion are involved in the formation of substitutions. As expected M28V (1a) owns a score of 1 (Table 1) that correlates with high percentage of substitution (57.9%). How to explain the A30S (3a) has a higher score (>2.5) and higher percentage up to 31.0% but the Y93H (3a) with a score of 1 (prone to mutate) but only 2.2% in nature?

6. How to explain the difference of CD8 epitopes in Russian and worldwide cohorts as shown in Figure 4b and 4d?

7. The discussion session is over-sized. It fails to inteprete the results and comprehensively compare the results with other literatures.

Author Response

General comment:

…Although massive analyses were performed, the authors messed up scientific roles of these substitutions in viral replication, potential virus resistance to the DAAs and viral immune escape. According to current in silico studies, the authentic and exact role of the substitutions however is ambiguous. Furthermore, it is not proper to define them as "resistance"-associated substitutions before DAA treatment, since DAA has not been administrated yet. They are mutations during replication but not substitutions passively driven by DAA treatment, although they showed the same mutative directions in comparison to identified RASs after treatment.

Comment 1.

RAS stands for resistance-associated substitution upon DAA treatment, which is distinct from viral mutation. RAS is a passive process, a selective effect by DAAs that kill replication of wild-type virus. However, mutation is active and spontaneous probably driven by viral replicative advantange or host immune pressure, however neither is associated with DAA treatment in treatment-näive patients. HCV could not predict a future DAA treatment and evolves RASs in advance. Also a major problem is that the word "DAAs" is present in article and abstract but only in Figure 1 and Table 1, other data are not directly related with DAA.

We are grateful to the Reviewer for the comments and thorough analysis of the manuscript.

This important issue was raised by the Editor. Hereby, we address it in more detail. As in the answer to Reviewer 2, we would like to stress that

amino acid residues in certain positions in HCV NS5A associated with resistance to DAA do not evolve as a result of treatment, but pre-exist among HCV variants, and get replicative advantages after DAA application. In our manuscript, we have shown that HCV variants with amino acid substitutions providing for resistance to DAA form a natural part of the viral population (page 13, lines 372-375). Our in silico evaluations indicate that some of them give the hosting virus an advantage of being poorly recognized by the immune system (weaker than HCV variants without these substitutions). Thus, some of these mutations are responsible not only for the escape from the action of DAA, but also for immune escape. We suggest this to be one of the mechanism(s) driving their appearance and spread, prior to DAA application.

In application to many viruses, for example HIV-1, the term “resistance-associated substitutions (mutations)” reflects mutations which do not naturally occur but do appear after the application of drugs. On contrary to this and possibly confusing, HCV virology uses the term “resistance-associated substitutions” to describe certain viral variants out of the multitude of natural quasi species which happen to be resistant to DAA and which are selected/acquire replication advantage after DAA application. This term is widely used in the clinical guidelines describing the role of HCV polymorphisms present at the baseline in the treatment-naïve patients (European Association for The Study of The Liver. "EASL recommendations on treatment of hepatitis C 2018." Journal of hepatology 69.2 (2018): 461-511) and also in the publications on the subject (see list of references below).

We called polymorphisms in NS5A responsible for resistance to DAA as “resistance associated mutations” (RAS) to adhere to this terminology and recommendations. Introduction contains now a sentence with the formal reasons motivating the use of this term (page 2, lines 57-60).

In response to Comment 1, indeed, we are not investigating viral evolution and appearance of substitutions associated with resistance to DAA as the result of DAA application. We study the landscape of RAS before the treatment. Hence, we do not focus on DAA, and they are mentioned only in the context of their ineffectiveness against NS5A variants carrying specific RAS (Fig 1, Table 1). Whereas RASs are present in the manuscript title, this term present in all figures/figure legends and tables, and throughout the manuscript text as the mechanism of their evolution and persistence is the main subject of this study.

References on the use of term “RAS”

1. Pawlotsky, Jean-Michel. "Hepatitis C virus resistance to direct-acting antiviral drugs in interferon-free regimens." Gastroenterology 151.1 (2016): 70-86.

2. Wyles, David L. "Resistance to DAAs: when to look and when it matters." Current HIV/AIDS Reports 14.6 (2017): 229-237.

3. Dietz, Julia, et al. "Consideration of viral resistance for optimization of direct antiviral therapy of hepatitis C virus genotype 1-infected patients." PloS one 10.8 (2015).

2.

I have to argue that as a RNA virus, HCV replication creates thousands of mutations in treatment-näive patients every day -in this regard- mutations on all viral proteins may be of importance, although NS5A is one of the most essential component but not the sole in viral replication. In this case, mutations (or substitutions) are either promoting viral replication or escaping immune recognition, again if you are talking about treatment-näive patients. However after DAA treatment RASs are pretty specific, since DAAs are direct-acting on one viral protein (for instance daclatasvir is a specific NS5A inhibitor) therefore only after treatment that targets one protein without affecting others we can say that the substitutions are resistance-associated. In sum, one virotype resided in the patient reflects a balance of viral replication and immune pressure, later on resistance becomes a topic only when treatment starts.

We absolutely agree with Reviewer on this issue. Our data demonstrate that such substitutions exist prior to application of DAA. DAA treatment acts as a selective force. Our study tackles different factors behind generation and survival/spread of these variants in treatment-naïve population, such as the ease of their spontaneous generation, replication competence of the resulting virus/functional adaption reflected by covariance networks, and  immunological adaptation/immune escape.  As we stress in the Discussion, our study supports the concept of drug resistant HCV variants as a natural part of viral population (page 13, lines 372-375).

3.

What are biological functions and fates of these characterized substitutions after DAA treatment? As expected wild-type virus is sensitive to NS5A inhibitor and virus with substitutions becomes dominant. Overall effect would be that those patients are not suitable for daclatasvir. Whether they are responders to other agents in the cocktail regimen however is largely unknown. It would be better to moniter patients after DAA treatment, measure virus polymorphism and more importantly present a concrete therapeutic plan specific for those patients.

The main goal of this study was to assess prevalence of NS5A RASs before the application of DAA. This is a “zero” point. The next step, as indicated in the Conclusion, would be to assess the prevalence and circulation of RAS-containing HCV strains in different ethnical groups after the introduction of DAAs into the clinical practice (page 15, lines 447-451).

The first combination of DAA was registered for use in the Russian Federation only in 2015. Since 2016, Russian Federation has programs providing free treatment with DAA to patients with chronic hepatitis C in six most affected regions. We are collected samples and data on the prevalence of RAS in NS5A in the regions included in this program. This study would reveal if certain NS5A RASs give HCV advantages and result in the preferential spread of the respective strains with diminishment of the spread of HCV strains without RAS. Interestingly, however, comparison of our data on the spread of HCV with RAS in NS5A in Russia prior to introduction of DAA with the data on their spread in countries widely using DAA shows no major differences (Krishnan, P. et al. Analysis of Hepatitis C Virus Genotype 1b Resistance Variants in Japanese Patients Treated with Paritaprevir-Ritonavir and Ombitasvir. Antimicrob Agents Chemother. 2015, 60(2), 1106-13, doi: 10.1128/AAC.02606-15; Lu, J. et al. Subtype-Specific Prevalence of Hepatitis C Virus NS5A Resistance Associated Substitutions in Mainland China. Front Microbiol. 2019, 10, 535, doi: 10.3389/fmicb.2019.00535; Hernandez, D. et al. Natural prevalence of NS5A polymorphisms in subjects infected with hepatitis C virus genotype 3 and their effects on the antiviral activity of NS5A inhibitors. J Clin Virol. 2013, 57(1), 13-8, doi: 10.1016/j.jcv.2012.12.020.). This indicates predominant role in HCV evolution and acquisition of substitutions conferring resistance to DAA of factors other than drug escape. We have tackled this in the conclusion as it strengthens the role in viral evolution of the factors addressed by our study (page 15, lines 444-447).

We have also discussed practical implementations of our data regarding screening for NS5A RASs at the baseline with consequent selection of optimal treatment options (page 12, lines 326-330).  

4.

Among NS5A substitutions Leu31 (L31) and Tyr93 (Y93) are well-characterized. Kohler et al. (2014) have presented a predicted structure of NS5A carrying the mutations and NS5A inhibitor. If M28V (genotype 1a) and A30S (3a) interfere with NS5A-inhibitor binding, such in silico structure is required.

Careful review of the literature reveals that most of RAS in NS5A lie outside of its functional motives and signatures. We have added this statement to the discussion (page 13, lines 340-357). This information makes prediction of possible involvement of NS5A RAS in protein functions hypothetical with gross limitation to sterical interference. We felt it to be insufficiently reliable for inclusion into the manuscript. Instead, we added citations to the experimental studies of the structure and function of NS5A in relation to regions of RAS (Discussion, page 13, lines 341-346) and of NS5A with RAS, including those of Kohler et al 2014, to Introduction (page 2, lines 78-80), and Bartlett et al. 2018 to the discussion (page 13, lines 345-349).  

5.

 The authors showed that both nucleotide transition and transversion are involved in the formation of substitutions. As expected M28V (1a) owns a score of 1 (Table 1) that correlates with high percentage of substitution (57.9%). How to explain the A30S (3a) has a higher score (>2.5) and higher percentage up to 31.0% but the Y93H (3a) with a score of 1 (prone to mutate) but only 2.2% in nature?

The high prevalence of HCV 3a A30S could be due to its introduction into population (please see genetic relatedness of the majority of HCV 3a sequences bearing A30S on phylogenetic tree, Supplementary Figure  S2b). Its preferential spread may be an outcome of the successful immune escape, as our data demonstrate the decrease of predicted CD8 T-TCR binding scores for peptides bearing RASs in positions 28-30 (but not in position 93). We added this point into Discussion (page 15, lines 434-437).

6.

How to explain the difference of CD8 epitopes in Russian and worldwide cohorts as shown in Figure 4b and 4d?

For the global sequence set, we demonstrated that introduction of RAS in the region between aa 28-32 of HCV 1b led to the decrease in the peptide binding scores by both CD8+ and CD4+ T cells. This data contrasts an increase in peptide binding scores observed for the Russian HCV 1b sequence set.  

The first possible explanation may be as follows:

Peptide binding scores of the global and for the Russian sets of HCV 1b sequences were determined based on the 10 most globally common HLA types. This is a logical approach for the analysis of the global HCV sequences. Analysis demonstrated that NS5A variants with RAS at 28-32 have lower TCR binding scores than variants without these substitutions. We applied same criteria in the analysis of TCR binding scores for the peptides harboring RAS of NS5A in the Russian HCV 1a, HCV 1b and HCV 3a. In most of the cases, we could show same trends for the Russian sequence sets as for the global sequences, namely, the decrease in the TCR binding scores, although in this specific case, for peptides covering aa 28-32, this was not the case. TCR peptide binding scores were higher for the variant with than for the variant without substitutions. We recorded this event although it did not support our concept of immune escape as driving force of NS5A evolution. However, one has to keep in mind that Russia is a huge country with hundreds of ethnicities, fine haplotyping of the population in the Russian Federation is far from completion, and currently show different prevalence profiles in different territories. This potentially results in a differential immune pressure with outcome depending on HLA restriction of the dominant T cell epitopes localized in each RAS region. We are currently elucidating it in a study of prevalence of specific RAS in NS5A in specific territories of the Russian Federation with available information on the locally prevalent HLA types. With a more precise selection of the most common HLA types, the results of the analysis of actual immune pressure on NS5A of HCV 1b in the territory of the Russian Federation could be different.

The second possible explanation for the difference between results obtained for Russian and “global” HCV 1b sequence sets might be due to bias in set formation: the subset of Russian HCV 1b sequences contained all RAS-bearing sequences, including those with substitutions in positions 28, 30, 31, 58, 62, 92, and 93), while the “global” HCV 1b RAS subset contained only Q30R.

This is a hypothetical explanation and we felt inappropriate and speculative to incorporate it into discussion. Instead, we limited ourselves to formulation of the necessity of such study in the Discussion (page 14, lines 425-430).

7.

The discussion session is over-sized. It fails to inteprete the results and comprehensively compare the results with other literatures.

We shortened the Discussion section. Following parts were removed: page 11, lines 246-252; page 11, lines 270-273; page 14, lines 382- 390; page 14, lines 400-401; page 15, lines 437-440. We also strengthened the links between our data and earlier published studies on RAS. We hope that in the revised form it presents a comprehensive comparison of the results of this study with the earlier studies.

Round 2

Reviewer 2 Report

In current version, the authors have addressed most of the comments, revised the manuscript and removed all redundant concepts in the discussion, which makes the story more suitable for consideration. Furthermore, existing data are acceptable. However, some scientific questions are still arguable.

  1. In response to authors' response to the application termed "resistance-associated substitutions (RASs)", I have read the guideline "EASL recommendations on treatment of Hepatitis C 2018". My understanding is that the definition of RASs is recommended to be used in two occasions: (1) for patients at baseline (as shown at Page 465) prior to treatment; (2) for patients previously exposed to DAAs prior to retreatment of other DAAs (as shown at Page 497). Since in the case (2) RASs are probably related to the DAA treatment that is distinct from the case (1), although they look "convergent". Hence, I recommend the authors to replace the "RESISTANCE-ASSOCIATED SUBSTITUTIONS IN HEPATITIS C VIRUS NS5A PROTEIN BEFORE THE INTRODUCTION OF DIRECT ACTING ANTIVIRALS" in the article title with "VIRAL QUASISPECIES IN HEPATITIS C VIRUS NS5A PROTEIN IN TREATMENT-NÄIVE PATIENTS" to avoid misunderstandings by readers. However, making clearer clarification in both cases, RASs can still be used in the abstract and other parts of the paper.
  2. It is also worth noting that observed existing single and double substitutions in the NS5A gene might be kinetic over time. It is difficult to predict whether they become resistant as non-responders to any DAAs.
  3. In this paper, more viral polymorphisms have been identified compared to the Table 9 in above EASL recommendation. Specifically, subtype/genotype 1b Q/E62K(1.1%)/S(1.1%) and 3a S62A(1.1%)/I(2.2%)/P(4.4%)/T(7.8%)/V(1.1%)/W(1.1%) natural mutations are novel and a plus significance of this work.

Author Response

Comment 1.

In response to authors' response to the application termed "resistance-associated substitutions (RASs)", I have read the guideline "EASL recommendations on treatment of Hepatitis C 2018". My understanding is that the definition of RASs is recommended to be used in two occasions: (1) for patients at baseline (as shown at Page 465) prior to treatment; (2) for patients previously exposed to DAAs prior to retreatment of other DAAs (as shown at Page 497). Since in the case (2) RASs are probably related to the DAA treatment that is distinct from the case (1), although they look "convergent". Hence, I recommend the authors to replace the "RESISTANCE-ASSOCIATED SUBSTITUTIONS IN HEPATITIS C VIRUS NS5A PROTEIN BEFORE THE INTRODUCTION OF DIRECT ACTING ANTIVIRALS" in the article title with "VIRAL QUASISPECIES IN HEPATITIS C VIRUS NS5A PROTEIN IN TREATMENT-NÄIVE PATIENTS" to avoid misunderstandings by readers. However, making clearer clarification in both cases, RASs can still be used in the abstract and other parts of the paper.

We are very grateful to Reviewer for comments and the raising the scientific issues.

We agree that the use of the same term “RAS” in two different occasions may lead to the misunderstanding. To avoid this, we changed the title of the manuscript to adhere to the Reviewer’s suggestion (page 1, lines 2-7) and added the clarification of the use of term “RAS” to the Introduction section (page 2, line 61-65).

Comment 2.

It is also worth noting that observed existing single and double substitutions in the NS5A gene might be kinetic over time. It is difficult to predict whether they become resistant as non-responders to any DAAs.

Indeed, the presence of these substitutions in NS5A does not necessarily mean that these patients would not respond to the therapy. Moreover, these variants might disappear from viral quasispecies over the time. We added the respective note to the Discussion (page 12, lines 331-333).   

Comment 3.

In this paper, more viral polymorphisms have been identified compared to the Table 9 in above EASL recommendation. Specifically, subtype/genotype 1b Q/E62K(1.1%)/S(1.1%) and 3a S62A(1.1%)/I(2.2%)/P(4.4%)/T(7.8%)/V(1.1%)/W(1.1%) natural mutations are novel and a plus significance of this work.

We are grateful to Reviewer for stressing the significance of these new data. We mentioned these substitutions, described their plausible roles in resistance to DAA, and stressed that their clinical significance has yet to be elucidated (page 12, lines 320-326).